# Randomized Personalized Trial for Stress Management Compared to Standard of Care

**DOI:** 10.3390/jpm16010023

**Published:** 2026-01-04

**Authors:** Ashley M. Goodwin, Thevaa Chandereng, Heejoon Ahn, Danielle Miller, Stefani Slotnick, Alexandra Perrin, Ying Kuen Cheung, Karina W. Davidson, Mark J. Butler

**Affiliations:** 1Northwell Health, New Hyde Park, NY 11042, USA; 2Institute of Health System Science, Feinstein Institutes for Medical Research, Northwell Health, Manhasset, NY 11030, USA; 3Mailman School of Public Health, Columbia University, New York, NY 10032, USA; 4Donald and Barbara Zucker School of Medicine at Hofstra/Northwell, Hempstead, NY 11549, USA

**Keywords:** personalized trials, N-of-1, stress management techniques, mindfulness, yoga, physical activity

## Abstract

**Background/Objectives**: Psychological stress is a common problem but hard to universally treat. Personalized (N-of-1) trials assess a participant’s response to multiple specific interventions. Though personalized (N-of-1) trials have been used in select interventions, no prior research has examined whether N-of-1 designs provide superior stress reduction relative to standard of care. **Methods**: Participants were randomized to personalized N-of-1 (N = 106) or standard-of-care (N = 106) arms for three stress-management interventions (mindfulness meditation; yoga; brisk walking). All participants completed ecological momentary assessments (EMA) of stress three times daily for 18 weeks (2-week baseline, 12-week intervention, 2-week assessment, and 2-week follow-up). After the intervention, participants in the N-of-1 arms received a personalized report identifying which intervention worked best for them. All participants chose one intervention to manage their stress during follow-up. The primary outcome was change in perceived stress between baseline and follow-up. **Results**: Participants in the personalized (N-of-1) arms did not report significantly reduced EMA stress levels relative to standard-of-care (*p* = 0.496), though the effect was stronger among N-of-1 participants who chose the stress-management intervention recommended by their report [B(SE) = −0.67(0.34); *p* = 0.049]. **Conclusions**: Results show the potential of personalized (N-of-1) trials to provide individuals with information unique to them to help identify interventions for stress management. However, many participants in the personalized trial arms did not choose the intervention recommended by their trial. Additional research is required to refine how personalized (N-of-1) trials are conducted and how trial results are reported to participants to ensure the maximal benefit of these trial designs.

## 1. Introduction

Psychological stress is a highly prevalent condition with significant impacts on mental and physical health [1]. Psychological stress has been associated with increased risks for cardiovascular disease [2,3,4], depression [5], reduced sleep efficiency [6], and poor health behaviors such as smoking [7].

Mindfulness meditation, yoga, and physical activity are stress management techniques associated with significant reductions in both physiological measures or self-reported measures of stress, though these commonly used techniques have heterogeneous treatment effects [8,9,10,11], indicating that not every intervention will provide uniform benefits for all patients. Even with these multiple, evidenced-based stress management treatments, little is known on which treatment will be most effective for stress reduction in an individual patient.

In cases where treatments have heterogeneous effects, personalized (N-of-1) trials may provide an opportunity for selecting the appropriate treatment for each individual [12,13,14]. Personalized (N-of-1) trials are a series of single-case, randomized, crossover trials where each individual is the unit of analysis [15]. Previous research has demonstrated the potential impact of personalized (N-of-1) trials for helping patients to identify how to manage diseases and symptoms [16]. For example, patients concerned about statin-related myalgias were more likely to resume statin treatment after taking part in personalized (N-of-1) trials which identified pain symptoms as nocebo effects [17,18].

Despite the potential benefits of personalized (N-of-1) trials, they are infrequently conducted and few have investigated the effectiveness of N-of-1 trials to improve clinical outcomes compared to standard-of-care [14,19]. Further, the decision to use personalized trials is often led by clinicians or researchers [20,21], with limited input from patients. Our approach differed in that our design included providing information uniquely relevant to the participant, thus allowing for the examination of the “match” between the treatment recommendation based on the personalized trial data and the treatment selected by an individual [22]. We hypothesized that providing access to individualized data on how well the stress management techniques worked would result in participants being more involved in the treatment decision, and thus more likely to choose the most effective treatment [23,24,25].

The aim of the current study was to determine if personalized (N-of-1) trial designs allow for improved stress management compared to standard of care. By utilizing a person-centered research design, we aimed to help 212 individuals who self-identified as experiencing elevated stress (Perceived Stress Scale, PSS) [26] utilize detailed personalized (N-of-1) trial data to inform how they manage their symptoms of stress. Further, we wanted to empirically evaluate the benefits of our personalized (N-of-1) trial approach versus traditional standard-of-care wherein individuals use whatever approach they feel works best. We hoped that by providing participants with rigorous, holistic personalized (N-of-1) data, we could allow participants in personalized (N-of-1) trials to select which stress management intervention was the most empirically effective approach for them and allow them to better manage their symptoms of stress.

## 2. Materials and Methods

### 2.1. Study Design

Full details of the current study’s design are available in our published protocol paper [22], and in part are summarized in this section. Briefly, the current study is a randomized, National Institutes of Health (NIH) Stage Model of Behavioral Intervention Stage II trial design examining the effect of personalized (N-of-1) trials relative to standard-of-care. We randomized 212 participants: 106 to a personalized (N-of-1) trial and 106 to standard-of-care (Figure 1).

As detailed in our protocol paper [22], the intervention was delivered virtually to participants residing in the United States over the course of 18 weeks, divided into a 2-week baseline period, a 12-week intervention period, a 2-week assessment period, and a 2-week post-intervention observation period (Appendix A). Participants’ levels of stress were assessed by ecological momentary assessment (EMA) measures [27,28,29]. A wearable activity tracker (Fitbit^TM^, San Francisco, CA, USA) was used to monitor steps continuously and sleep. Intervention components were delivered by virtual link to an online video or audio recorded by an experienced Zeel wellness provider [22].

### 2.2. Participants, Recruitment and Consent

As detailed in our protocol paper [22], participants were recruited digitally from across the United States. One recruitment focus included outreach to ~104,000 employees of the Northwell Health System. Within-system online recruitment methods included e-mail blasts to Northwell employees and postings on internal employee media channels (i.e., Yammer). Internet and social media postings beyond Northwell were conducted via Facebook and Google advertisements [22]. Additionally, online posts to communities on platforms such as Reddit, Craigslist, and LinkedIn were used to appeal to individuals with elevated stress. To be eligible for the study, participants were required to be ≥18 years old and have a Perceived Stress Scale (PSS) score of 20 or higher [26,30]. Full eligibility criteria can be found in Appendix A. Recruitment materials directed individuals to an online webpage with details about the study and digital screening measures containing questions regarding study inclusion and exclusion criteria [22]. If study inclusion criteria were met, eligible individuals were provided with a link to consent. Consent was obtained electronically via REDCap with a short video explaining details of the study protocol and consent form. This study was approved by the Northwell Health Institutional Review Board (IRB) [22].

### 2.3. Baseline Period

As detailed in our protocol paper [22], the first two weeks of the study was a baseline assessment period. Participants were asked to engage in their usual methods of managing stress and were instructed to wear their Fitbit device at all times, including during sleep. Participants also completed EMA ratings of their stress, fatigue, confidence, concentration, mood and pain three times daily at random times via text message, and a short daily survey assessing whether they did anything to manage their stress that day and if they experienced any side effects in doing so [22]. Each week, participants also completed the Perceived Stress Scale. Participants were asked to wear their Fitbit devices day and night (>12 h/day), to sync their device with the Fitbit application on their phone at least every two days and charge their Fitbit device at least every four days. Participant adherence, defined by 720 min (12 h) of Fitbit wear time on 80% of days and 80% EMA survey adherence during baseline, was assessed prior to randomization to the intervention [22].

### 2.4. Intervention Period

#### 2.4.1. Stress Management Techniques

As detailed in our protocol paper [22], three stress management techniques commonly used were included in all arms: guided mindfulness meditation, guided yoga, and guided brisk walking. Zeel, a commercial wellness technology platform that connects individuals to in-home or in-office services (such as massage and yoga), was contracted to create recorded guided video and audio content [31]. Recorded video and audio stress management content was stored on a commercial website, Vimeo [22]. Data collected included the date the video was viewed and corresponding view duration. Participants were able to access intervention videos through smartphone or desktop devices [22].

#### 2.4.2. Randomization

As detailed in our protocol paper [22], of those participants who were enrolled in the study, 53 were randomized into Arm 1, 53 were randomized to Arm 2, and 106 were randomized to Arm 3 by the study statistician. Arms 1 and 2 received two treatment orders of Mindfulness Meditation, Yoga, and Brisk Walking in 6, 2-week blocks [22]. Participants in Arm 3 received standard-of-care where they were allowed to engage with treatments on their own self-guided schedule (Appendix A). Prior to the first participant enrollment, the study statistician and data team generated a permuted block randomization sequence in 27 blocks of 8 participants with pre-specified ratios to ensure equal treatment allocation within each block [22]. As participants enrolled, they were assigned to arms sequentially using this randomization sequence [22]. Participants, study staff, and statisticians were not blinded to treatment following assignment.

#### 2.4.3. Intervention

As detailed in our protocol paper [22], during the intervention period, participants in the personalized trial arms (Arms 1 and 2) were provided with an intervention schedule for when they would complete stress management interventions. Participants in standard-of-care (Arm 3) were allowed to use the guided stress management interventions at their own pace. Participants were instructed to access their intervention video(s) through their unique folder on Vimeo. During intervention weeks, all participants received weekly evening reminders to complete their interventions along with a link and password to access the videos [22]. Participants in Arms 1 and 2 were able to complete their assigned intervention three times in the Monday-to-Sunday week for a total of 12 times per intervention and 36 total sessions [22]. They also received a morning reminder three times per week on their preferred days as indicated in their onboarding survey to complete their interventions along with a link and password to access the video [22]. Participants in Arm 3 self-directedly completed the 3 interventions no more than 12 times each for a total of 36 sessions (same total amount of intervention sessions as Arm 1 and 2 participants) [22]. All participants were asked to refrain from engaging in other stress management interventions outside their usual regimen for the duration of the study. During all intervention periods, participants were asked to wear the Fitbit device 24 h a day and answer the four EMA survey measures and daily survey sent to them via text message each day.

#### 2.4.4. Participant Report

As detailed in our protocol paper [22], after the intervention period, participants’ data was analyzed by statisticians and presented in personalized data reports. The report compares baseline measurements of activity, sleep, concentration, confidence, mood, pain, fatigue, and stress to intervention period measurements [22]. Reports for participants in Arms 1 and 2 included information about which of the three interventions provided the most reduction in momentary stress as well as information about how various other outcomes (e.g., physical activity, sleep, fatigue, pain) changed during weeks the participant was asked to complete each intervention [22]. Reports for participants in Arm 3 provided details about how stress and other outcomes changed over time but were not provided information about how each of the three interventions may have impacted stress [22].

#### 2.4.5. Post-Intervention Observation Period

As detailed in our protocol paper [22], once participants received their participant reports, regardless of Arm, they were asked to select an intervention to continue receiving for an additional two weeks. Based on their selection, participants received another 6 sessions of stress management content (e.g., if yoga was selected, participants received 6 additional views of the yoga content); however, they were not prompted to complete the selected intervention. Participants were observed for 2 weeks and were asked to continue to wear their Fitbit and answer study surveys. At the end of the 18 weeks, each participant was provided with a satisfaction survey. Upon completion of data monitoring, participants were given instructions on how to un-link their Fitbit from the study account [22]. Study recruitment began in May 2022, and trial completion occurred in January 2024.

### 2.5. Primary Outcome

The primary outcome is the between-arm change in average daily perceived stress from baseline to follow-up, assessed using ecological momentary assessment (EMA) [32]. EMA stress was assessed using a single item asking participants to rate their stress in the current moment on a scale of 0 to 10. This measure was adapted from the Numeric Pain Rating Scale [33] and has previously been utilized as a self-reported stress measure in other personalized (N-of-1) trials [34,35,36]. Participants were prompted to respond to a question about their stress “Please answer the following questions according to how you’re feeling at this time” and answer on a scale of (0) Not at all to (10) Very much, with (1) to (3) being A little bit, (4) to (6) being Somewhat, and (7) to (9) being Quite a bit [22]. The timing of the text messages was randomized between a participant’s self-reported wake and sleep times. Changes in EMA stress are reported by the mean difference between average stress levels during baseline and average stress levels during follow-up [22].

### 2.6. Secondary Outcomes

The major secondary outcome is proportion of times participants selected the intervention which was recommended to them via their personalized data report. At the end of the intervention in the personalized trial arms (Arm 1 and Arm 2), personalized trial data was used to identify during which intervention (mindfulness, yoga, or walking) the most stress reduction occurred [22]. We identified the proportion of participants who selected the intervention recommended by their report and those who did not. Participants in Arm 3 were similarly provided with a personalized report and asked to choose a preferred intervention; however, this report did not provide a recommended intervention [22].

Other secondary outcomes include weekly perceived stress assessed with the 10-item Perceived Stress Scale (PSS-10), modified to be delivered to assess the prior week rather than the prior month [22]. The minimum total score possible is 0 and the maximum total score possible is 40. Higher values represent higher levels of stress. The PSS-10 can be found in the appendix. Daily self-reported fatigue, pain, concentration, mood, and confidence ratings were also assessed via EMA using single items with a similar format to EMA stress, as described above. For example, for “I feel fatigued”, participants responded by selecting (0) Not at all, (1) to (3) being A little bit, (4) to (6) being Somewhat, and (7) to (9) being Quite a bit, or (10) Very much [22]. Delivery and assessment of these measures were identical to the methods used for EMA stress described above.

As detailed in our protocol paper [22], feasibility was measured by the mean usability score via the 10-item System Usability Scale (SUS) [37] The SUS is a validated questionnaire that asks users to score each item on a Likert scale from Strongly Disagree (1) to Strongly Agree (5). Individual item scores are multiplied by 2.5 and summed to generate a total score ranging from 0 to 100, with higher scores indicating a greater level of usability. This measure has been utilized and validated in multiple contexts [37,38]. The SUS was presented to participants as addressing the ease of use, complexity, and consistency of the personalized trials system as a whole and can be found in the appendix.

Satisfaction was measured using a 13-item satisfaction survey. The survey assessed participant satisfaction with elements of the trial, including resources like the Fitbit device, the N-of-1 trial design, survey assessment measures, interventions, and the participant report. Participants were asked to rate their satisfaction on a scale of 1 “Not very satisfied” to 5 “Very satisfied.” The satisfaction survey included a 7-item series of questions regarding the participants’ experience in the personalized trial as a whole [22]. Satisfaction with aspects of the trial including the onboarding process and ease of trial adoption are rated on a scale of 1 “Strongly Disagree” to 5 “Strongly Agree” [22]. Finally, satisfaction with the personalized report was assessed through 5 questions on a scale of 1 “Strongly Disagree” to 5 “Strongly Agree” [22]. The final 3 items on the satisfaction survey ask if the participant would recommend the personalized trial to others with stress (“I would not recommend” to “I would strongly recommend”), how helpful participating in the study was in regard to their symptoms of stress (“Not at all helpful” to “Extremely helpful”), and a free-text comments box [22]. The satisfaction survey can be found in the appendix.

### 2.7. Analysis

#### 2.7.1. Sample Size Calculation

As detailed in our protocol paper [22], the sample size was calculated based on the primary outcome, change in average momentary stress between the baseline and follow-up periods. Estimates of the sample were generated using a two-sample *t*-test comparing the participants in the personalized trial arms versus the participants in the standard-of-care arm. With an effect size of Cohen’s d = 0.307 identified using pilot data from a previous N-of-1 trial [35] and an alpha level of 0.05, the current trial will achieve 80% power with a sample size of 168 participants (84 in the personalized arms and 84 in the standard-of-care arm) [22]. To ensure sufficient sample size for the primary outcome, we assumed a conservative 20% attrition rate over the course of the trial, yielding a final sample of 212 participants (106 in the personalized arms and 106 in standard-of-care) [22].

#### 2.7.2. Primary Analysis

As detailed in our protocol paper [22], to determine whether the personalized intervention (Arms 1 and 2) yield greater reductions in stress relative to standard-of-care (Arm 3), changes in the primary outcome (EMA stress) were examined between baseline and follow-up periods first using a two-sample *t*-test. This test compares mean changes in stress between baseline and follow-up by treatment condition. We hypothesized that participants in the personalized arms would demonstrate greater stress reductions than participants in standard-of-care [22]. To examine how personalized interventions influenced stress over the duration of the trial, changes in EMA stress over time were compared between the personalized and standard-of-care arms using linear mixed effects (LME) with autoregressive models with order 1 (AR(1)) to account for linear trends between stress ratings over time [22]. This approach accounts for the serial autocorrelation common to personalized (N-of-1) trials [39]. Treatment condition (personalized versus standard-of-care) and week of the trial were utilized as fixed effects in the analysis and a random effect were specified for participants. Analysis for the primary outcome will be based on intention-to-treat (ITT) principle that includes all randomized participants, regardless of their level of treatment received or protocol adherence. Analyses of the primary outcome used all available data during follow-up. Participants who were missing all data during the follow-up period were excluded from these analyses. We explored using multiple techniques to account for missing values (including nearest neighbor interpolation and multivariate imputation by chained equations) to deal with missing data for participants who had no data during follow-up. We plan to explore the validity of these differing approaches in future research but decided to use all available data for the current analysis.

#### 2.7.3. Secondary Analyses

The proportion of participants in Arms 1 and 2 who selected the treatment recommended for stress reduction in their personalized report is represented with frequencies and percentages.

Participant responses to the SUS and rating of satisfaction are reported with means, standard deviations, and frequencies for participant responses for survey items [37]. As the SUS is a standardized measure, overall participant scores are compared to other comparable digital interventions and to previous digital N-of-1 trials.

As detailed in our protocol paper [22], additional outcomes including stress measured using PSS-10 and EMA measures of fatigue, pain, mood, confidence, and concentration were examined using linear mixed effects (LME) autoregressive models with order 1 (AR(1)) to account for linear trends between stress ratings over time. As with analyses of the primary outcome, time (in weeks) and treatment condition were set as fixed effect while participant will be used as a random effect in the model [22]. These analyses examine the effects personalized treatment and time on each outcome while accounting for individual differences in participants. These analyses were exploratory and results from these analyses should be interpreted cautiously due to potential bias from these multiple comparisons.

## 3. Results

### 3.1. Enrollment and Sample Characteristics

Of the planned 212 participants there were 53 randomized to personalized Arm 1, 53 to personalized Arm 2, and 106 to standard-of-care. Characteristics of the enrolled sample can be found in Table 1. In this trial, the mean age was 41.1 (SD = 12.6) years, sex was 72.0% female (n = 152), 57.3% White (n = 121), and 25.6% were Hispanic (n = 54). Baseline PSS score was 25.0 (SD = 3.9) and baseline EMA stress was 4.1 (SD = 1.7). Participant characteristics did not differ between the two treatment personalized arms and standard of care Arm 3, including baseline PSS and EMA stress scores (Table 1). Of the 212 participants enrolled, there were 155 participants (73.6%) who continued through the trial until follow-up and completed the system usability scale (SUS) after trial completion [81 in the personalized Arms 1 and 2 (76.4%) and 74 in Arm 3 (69.8%)].

### 3.2. Perceived Stress and EMA Measures

The primary outcome results showed greater reductions in EMA stress over the course of the trial for the personalized arms [Mean (SD) = −0.50 (1.8)] compared to standard-of-care [Mean (SD) = −0.31 (1.7)], but these differences were not statistically significant (*p* = 0.496; Table 2). Changes in self-reported weekly stress were also not significantly different between personalized and standard-of-care arms (*p* = 0.780). In addition, there were no differences between baseline and follow-up for EMA pain (*p* = 0.397), EMA fatigue (*p* = 0.878), EMA mood (*p* = 0.856), EMA confidence (*p* = 0.100), and EMA concentration (*p* = 0.346).

The primary outcome results showed greater reductions in EMA stress over the course of the trial for the personalized arms [Mean (SD) = −0.50 (1.8)] compared to standard-of-care [Mean (SD) = −0.31 (1.7)], but these differences were not statistically significant (*p* = 0.496; Table 2). Changes in self-reported weekly stress were also not significantly different between personalized and standard-of-care arms (*p* = 0.780). In addition, there were no differences between baseline and follow-up for EMA pain (*p* = 0.397), EMA fatigue (*p* = 0.878), EMA mood (*p* = 0.856), EMA confidence (*p* = 0.100), and EMA concentration (*p* = 0.346).

### 3.3. Stress Management Interventions

We also examined which interventions were most commonly selected to manage stress during follow-up. In general, participants showed a preference for using brisk walking to manage stress during follow-up in Arm 1 [N (%) = 24 (55.8%)], Arm 2 [N (%) = 18 (45.0%)], and Arm 3 [N (%) = 37 (50.7%); Table 3]. Preference for which intervention was selected during follow-up did not differ between the personalized (N-of-1) arms and the standard-of-care arm (Pearson Chi-square *p*-value = 0.651, Table 3).

### 3.4. Recommendation for Stress Management

Among participants who completed the personalized trial arms and selected a stress management intervention to use during follow-up (N = 81), the concordance between selected stress management intervention with the recommended intervention per their personalized trial report was statistically significant (kappa = 0.24, *p* = 0.003). The personalized trial reports recommended mindfulness for 28 individuals (34.6%), yoga for 20 individuals (24.7%), and brisk walking for 33 individuals (40.7%; Table 4). Of the 28 individuals who were recommended mindfulness for stress management, only 12 (42.9%) chose to use mindfulness to manage their stress in follow-up. Of the 20 individuals who were recommended yoga, only 8 (40.0%) chose to use yoga during follow-up. Brisk walking was recommended for 33 individuals but was chosen by 21 (63.6%) of individuals in follow-up. Overall, only 41 (50.6%) of participants in the personalized trial arm followed the recommendation for stress management interventions (Table 4).

### 3.5. Sensitivity Analysis—Comparison of Self-Report Measures by Compliance with Personalized Trial Recommendations

Our findings suggest that though everyone in the personalized trial arms received unique feedback about which stress management intervention was most useful for reducing stress, only half of participants chose to comply with these recommendations during follow-up. To examine whether compliance with personalized trial recommendations influenced self-reported ratings of stress and other EMA measures, we compared participants who complied with recommendations and individuals who did not versus standard-of-care. We utilized linear regressions comparing compliers, non-compliers, and standard-of-care for changes in EMA measures and weekly stress between baseline and follow-up. These regression models were adjusted for participant age, sex, and race/ethnicity. Relative to standard-of-care, individuals who selected stress management interventions in accordance with their personalized trial results showed significant reduction in EMA stress relative to standard-of-care [B (SE) = −0.66 (0.34), *p* = 0.049; Table 5]. In contrast, individuals in the personalized arms who did not select their recommended stress management intervention had no change in EMA stress relative to standard-of-care [B (SE) = 0.08 (0.04), *p* = 0.825; Table 5]. All other regression comparisons for weekly stress and EMA measures were not statistically significant (*p* > 0.05).

### 3.6. System Usability Scale (SUS) and Participant Satisfaction

Participant ratings for the usability of the stress management trial were high overall [SUS Mean (SD) = 81.3 (14.0); Table 6] and in the range considered to be “excellent” for the SUS. Ratings of usability were high in the personalized trial arms [SUS Mean (SD) = 82.7 (13.0)] relative to the standard-of-care arm [SUS Mean (SD) = 79.8 (14.9)] but did not differ significantly between arms (independent-samples *t*-test *p*-value= 0.20; Table 6). Ratings for individual items on the SUS showed that the lowest rated item was “I think that I would like to use this system frequently” [Mean (SD) = 2.59 (1.13); Table 6], with 51% of participants saying they agree with this statement (Figure 2). The highest rated item was “I do not think that I would need the support of a technical person to be able to use this system” [Mean (SD) = 3.70 (0.69); Table 6], with 93% of participants saying they agree with this statement (Figure 2). Participant ratings for individual items on the SUS did not significantly differ between personalized and standard-of-care arms (*p* > 0.05; Table 6; Figure 2).

## 4. Discussion

This study compared a personalized (N-of-1) trial approach using three different stress management interventions to improve individual self-report of perceived stress (guided mindfulness meditation, guided yoga, and guided brisk walking) compared with a standard-of-care approach using the same three stress management interventions. The study shows mixed results for the benefits of personalized (N-of-1) trials for stress management. Though stress levels were lower during follow-up for individuals randomized to the personalized (N-of-1) trial arms relative to standard-of-care, this difference was not statistically significant at the 0.05 level of alpha. Surprisingly, this could mean that personalized (N-of-1) trials do not improve stress management. However, participants who selected the intervention recommended from their personalized (N-of-1) trial did show significant stress reductions relative to standard-of-care. This suggests that personalized (N-of-1) trials may be beneficial for participants but only for the sub-sample of individuals who followed the recommendations in their personalized report. However, these significant findings were identified in sensitivity analyses which were not specified a priori, suggesting these results should be interpreted cautiously. This sensitivity analysis may be biased.

Notably, participants in both the personalized (N-of-1) trial arms and standard-of-care arm had comparable ratings of usability and satisfaction with the trial. In addition, participant SUS scores for this series of personalized trials were comparable to previous series of personalized (N-of-1) trials conducted by the research team [34,35,36]. In addition, participant SUS scores were comparable or greater than other recent online stress management programs [40,41,42,43,44]. This is in contrast to the notion that personalized trials are burdensome for participants [45,46,47]. In fact, our results suggest that even though personalized (N-of-1) trials are more prescriptive in how interventions are used and delivered, the experience of participants is comparable between regimented intervention delivery and ad hoc intervention use during standard-of-care. The finding that either intervention is acceptable may mean that generalized advice for stress management is preferred for some while personalized recommendation is preferred by others.

Utilizing the structure and measurement of a personalized (N-of-1) trial, we were able to provide individuals with detailed and thorough feedback on how mindfulness meditation, yoga, and brisk walking helped to manage their levels of stress. Further, we were able to generate personalized reports with recommendations about which intervention would be most useful as a stress-management technique in the future [48]. Still, perceived stress did not differ between personalized (N-of-1) trial participants and standard-of-care participants. Similar results have been found in other trials comparing personalized trials to standard-of-care. For example, in a study to determine whether patients randomized to participate in an N-of-1 trial supported by a mobile health (mHealth) app would experience less pain and improved global health, adherence, satisfaction, and shared decision making compared with patients assigned to usual care, there was no difference between the N-of-1 and usual care groups [49]. Additionally, some have found in reviewing N-of-1 trial effectiveness that a notable proportion (38%) of personalized trials were indeterminate as to which treatment was best, with neither treatment being a clear “winner”, suggesting that the treatments are equally effective [21]. Some have suggested that in N-of-1 studies, even a non-significant effect between treatments does not necessarily mean there is no clinically meaningful effect [50]. In our study, we believe that a difference in stress between the N-of-1 arms and the standard-of-care arm may have been significant if a higher proportion of participants had followed their personalized recommendation for stress management intervention, a hypothesis that is supported by our sensitivity analyses. Another potential explanation for the limited effect of personalization on stress management that requires further consideration is the potential impact of the intervention materials (e.g., video content, and whether it was engaging, easy to use, or sufficiently motivating). While the focus of this trial was not on the intervention components themselves, and the study received high satisfaction ratings, future work could explore participants’ interaction with intervention components. More discrete fidelity measures regarding compliance with the intervention (i.e., beyond video tracking) may also boost the effect of personalization on stress management.

It should also be noted that there was differential loss to follow-up between the personalized and standard-of-care arms. In the personalized arms, 79% of participants selected their preferred stress management intervention during follow-up compared to 68% in standard-of-care. This suggests that individuals in the personalized arms may have been more motivated to continue through with follow-up. However, in other personalized (N-of-1) trials conducted by our group, the completion rates were 67% [35], 88% [34], 95% [36], respectively. Notably, the trial with the lowest completion level was disrupted by the COVID-19 pandemic [35]. The results of this study are in agreement with a review that reported that the completion proportion of N-of-1 trials was 80%, which was not meaningfully different from that of many group-level trials [21].

Another interesting finding was that there appeared to be a distinct preference in the trial for selecting brisk walking as a stress management intervention of choice. Roughly 50% of the sample chose to use brisk walking to manage their stress during follow-up. This overall preference may have contributed to participants choosing to ignore the recommendation from the personalized trial report and pick the stress management intervention they preferred. It may also indicate that these participants were already physically active and selected the intervention that best fit into their daily routine. It is also possible that participants found brisk walking to be more pleasant and chose that to be their stress management intervention regardless of the contents of their personalized trial report.

Only half of the participants in the personalized trial arms chose to use the treatment recommended by their personalized report during follow-up. This was unexpected, particularly in light of the high satisfaction rates. This rate of selecting the recommended treatment was lower than expected and is lower than the rates of compliance with personalized trial results shown in systematic review [21]. These results are also in contrast with a series of personalized trials that led to valuable changes in treatment, cessation of treatment, or confirmation of the original treatment [18,51,52,53] based on data unique to the patient. For example, of 71 N-of-1 trials for patients with any chronic pain, 65% of patients chose to change their pain medication due to trial results [54]. Further in contrast, in a personalized trial compared to usual care for treatment of chronic pain, shared decision making was significantly better in the N-of-1 group [49]. It is possible that personalized (N-of-1) trials may only be beneficial for individuals willing to act on personalized feedback. However, it is also possible that the design of our personalized trial reports presented too much information beyond self-reported stress, including other EMA measures, physical activity, and sleep. Our goal in providing this information was to allow participants to be fully informed about how each intervention affected them in a holistic manner. However, this breadth of information may have reduced the power of our recommendations for stress management. Studies have shown that patients prefer more information [55] and appreciate the opportunity to be involved in medical decisions [56]. It is also possible that providing additional information beyond a treatment recommendation can create information overload and a sense of being overwhelmed [57] that results in information avoidance [58]. Still, few RCTs have measures of participation in decision making and at least one health outcome [59], so our participation measures contribute to addressing this need.

### Strengths and Limitations

The primary strength of this study is that it is the first trial to compare personalized (N-of-1) trials for stress management to standard-of-care. Our approach to conducting personalized trials has been refined across multiple series of personalized trials, ensuring high levels of quality in our trial design, recruitment, outcome assessment, and intervention delivery [34,35,36,60]. To improve the validity of our comparison, participants in the standard-of-care arm received participant reports and data with comparable formatting and content to participants in the personalized trial arms.

The major limitation of the current study is that we did not anticipate only half of individuals enrolled in the personalized (N-of-1) trial arms would follow the stress management recommendations in their personalized report. Had we anticipated this, we could have specified a priori sensitivity analyses to evaluate this problem. Further, we could have conducted a thorough follow-up intervention to evaluate how and why participants selected stress management interventions during follow-up. Additional research will be required to evaluate how participants use the information in personalized trial reports to inform their decision making. Another potential limitation is that individuals may have had different sources and causes of stress. Although perceived stress is a complex and dynamic phenomenon influenced by both modifiable (e.g., coping skills) and non-modifiable (e.g., environment) factors [61,62], to make the content and the design of the intervention broadly applicable, we did not identify causes of stress or exclude individuals with particular causes of stress. However, this may have resulted in enrolling some participants with variable or potentially non-modifiable causes of stress. Because different causes of stress may respond differently to interventions, this is an important consideration for future work. Relatedly, while the EMA stress scale is a validated measure [63,64,65], that is also well-suited to a fully remote digital trial and has previously been utilized as a self-reported stress measure in other personalized (N-of-1) trials, we acknowledge that a 0–10 single-item stress scale may have limited reliability and sensitivity, especially for detecting between-group differences. A single item was chosen to reduce the potential participant burden caused by responding to EMA for 6 measures (stress, pain, fatigue, mood, confidence, concentration) 3 times daily. However, in reducing burden for participants and ensuring compatibility with previous series of N-of-1 trials, we utilized a less rigorous measure of our primary outcome. Lastly, one of our recruitment strategies primarily focused on health system employees, which may limit generalizability.

## 5. Conclusions

The findings for the current trial show that there is potential for personalized trials to help individuals select the proper interventions to manage their stress. In clinical practice, the nature of this N-of-1 trial design, leveraging digital recruitment and pre-recorded interventions, demonstrates strong potential for scalable N-of-1 personalized stress management. High participant satisfaction and the use of automated data collection suggest operational feasibility, with future cost reduction possible through further automation of report generation and leveraging patient-owned devices. Widespread clinical adoption will require integrating N-of-1 processes into routine workflows, ensuring clinician engagement, and improving patient uptake of personalized recommendations. Additional research is required to understand how individuals utilize the results of personalized trials to inform their treatment decisions and to refine how these results are communicated.

## Figures and Tables

**Figure 1 jpm-16-00023-f001:**
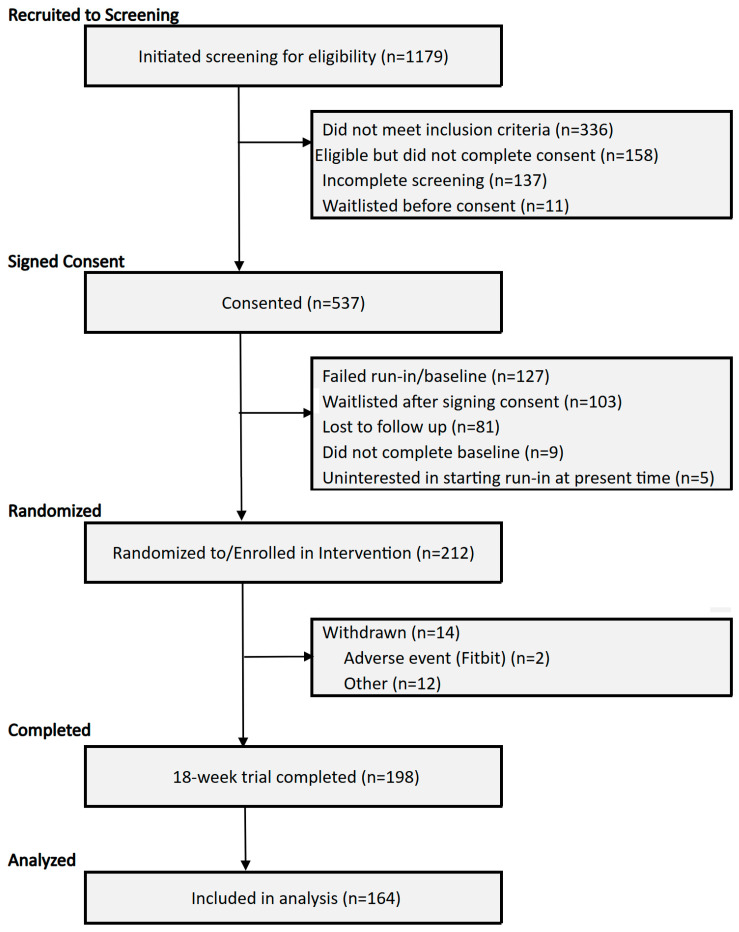
Participant CONSORT Diagram.

**Figure 2 jpm-16-00023-f002:**
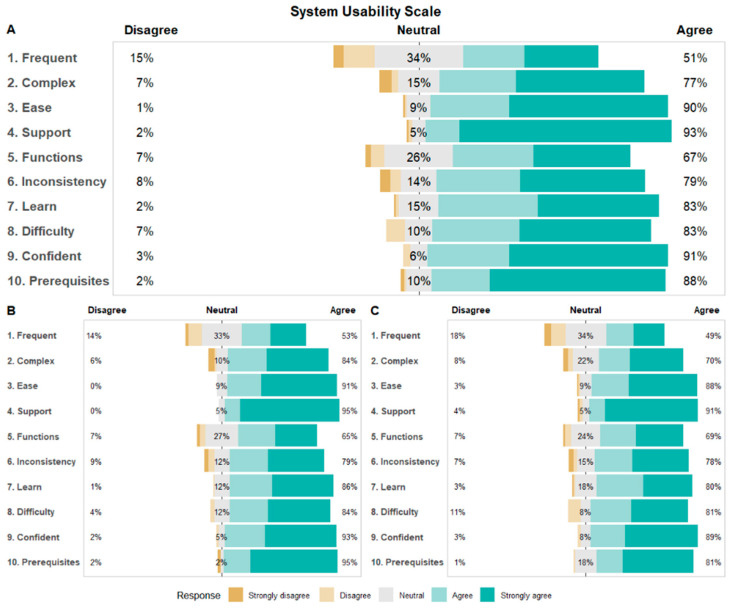
Participant System Usability Scale Ratings for Each Item Overall and By Arm. (**A**). Overall SUS likert plot. (**B**). Personalized Trial Arm SUS likert plot. (**C**). Standard-of-care Arm SUS likert plot.

**Table 1 jpm-16-00023-t001:** Baseline characteristics of the sample (N = 212).

Variable		Total Sample	Personalized Arm 1 ^a^ (N = 53)	Personalized Arm 2 ^b^ (N = 53)	Standard-of-Care Arm 3 (N = 106)	*p* Value ^c^
Age (years), mean (SD)		41.1 (12.4)	42.1 (12.2)	40.0 (12.9)	41.1 (12.3)	0.686
Gender, n (%)						0.608
	Female	154 (72.6%)	39 (73.5%)	42 (79.2%)	73 (68.9%)	
	Male	57 (26.9%)	14 (26.4%)	11 (20.8%)	32 (30.2%)	
	Other	1 (0.5%)	0 (0.0%)	0 (0.0%)	1 (0.9%)	
Race, n (%)						0.314
	Asian	31 (14.6%)	7 (13.2%)	7 (13.2%)	17 (16.0%)	
	Black	29 (13.7%)	9 (17.0%)	5 (9.4%)	15 (14.2%)	
	Mixed	8 (3.8%)	1 (1.9%)	5 (9.4%)	2 (1.9%)	
	Other	19 (9.0%)	4 (7.5%)	4 (7.5%)	11 (10.4%)	
	White	122 (57.5%)	30 (56.6%)	31 (58.5%)	61 (57.5%)	
	Declined/Unknown	3 (1.4%)	2 (3.8%)	1 (1.9%)	0 (0.0%)	
Ethnicity, n (%)						0.468
	Hispanic	36 (17.0%)	6 (11.3%)	12 (22.6%)	18 (17.0%)	
	Non-Hispanic	174 (82.1%)	47 (88.7%)	40 (75.5%)	87 (82.1%)	
	Unknown	2 (0.9%)	0 (0.0%)	1 (1.9%)	1 (0.9%)	
Perceived Stress Scale (PSS) Score, mean (SD)		25.0 (3.9)	24.4 (3.5)	25.7 (3.7)	25.0 (4.1)	0.986
Baseline EMA Stress, mean (SD)		4.1 (1.7)	4.3 (2.0)	4.3 (1.6)	4.0 (1.6)	0.101

^a^ Treatment order for Arm 1: Mindfulness meditation, Yoga, Brisk walking, Brisk walking, Yoga, Mindfulness Meditation. ^b^ Treatment order for Arm 2: Brisk Walking, Yoga, Mindfulness meditation, Mindfulness Meditation, Yoga, Brisk walking. ^c^ *p* values for comparisons of participant characteristics between treatment orders were obtained using independent-samples *t*-tests for age, PSS, and EMA stress scores and Chi-squared tests for all other characteristics.

**Table 2 jpm-16-00023-t002:** Change in Stress and Other Outcomes between Follow-up and Baseline.

Outcome	Change Between Follow-Up and Baseline Periods; Mean (SD)		Welch’s Two-Sample *t*-Test *p*-Value
	Personalized (Arms 1 and 2); N = 84	Standard-of-Care (Arm 3); N = 80	
EMA Stress	−0.50 (1.8)	−0.31 (1.7)	0.496
Weekly Stress *	−0.12 (0.3)	−0.14 (0.3)	0.780
EMA Pain	−0.09 (1.6)	0.10 (1.2)	0.397
EMA Fatigue	−0.73 (1.8)	−0.68 (1.7)	0.878
EMA Mood	0.58 (1.4)	0.54 (1.3)	0.856
EMA Confidence	0.35 (1.3)	0.68 (1.3)	0.100
EMA Concentration	0.26 (1.5)	0.48 (1.4)	0.346

* Smaller sample sizes existed for this comparison N = 69 (personalized) and 60 (standard).

**Table 3 jpm-16-00023-t003:** Selected Stress Management Interventions by Trial Arm and Recommended Stress Management Interventions in Personalized Trial Arms.

Selected Intervention	Personalized Trials; N(%)		Standard of Care; N(%)
	Arm 1 (N = 42)	Arm 2 (N = 39)	Arm 3 (N = 73)
Mindfulness meditation	9 (21.4%)	14 (35.9%)	23 (31.5%)
Yoga	10 (23.8%)	8 (20.5%)	13 (17.81%)
Brisk walking	23 (54.8%)	17 (43.6%)	37 (50.7%)
Recommended Intervention			
Mindfulness meditation	18 (42.9%)	10 (25.6%)	N/A
Yoga	6 (14.3%)	14 (35.9%)	
Brisk walking	18 (42.9%)	15 (38.5%)	

**Table 4 jpm-16-00023-t004:** Recommended Stress Management Interventions and Selected Stress Management Interventions in the Personalized Trial Arms (Arm 1 and Arm 2).

Selected Intervention	Recommended Intervention		
	Mindfulness Meditation (N = 28)	Yoga (N = 20)	Brisk Walking (N = 33)
Mindfulness meditation (N = 23)	12 (42.9%)	4 (20.0%)	7 (21.2%)
Yoga (N = 18)	5 (17.9%)	8 (40.0%)	5 (15.1%)
Brisk walking (N = 40)	11 (39.3%)	8 (40.0%)	21 (63.6%)

**Table 5 jpm-16-00023-t005:** Linear mixed effects (LME) regression examining the effect of complying with personalized trial recommendations.

Outcome	Standard-of-Care	Personalized Trial Non-Complier	*p*-Value	Personalized Trial Complier	*p*-Value
EMA Stress	REF	0.08 (0.035)	0.825	−0.66 (0.34)	0.049
Weekly Stress	REF	−0.02 (0.07)	0.787	−0.04 (0.07)	0.546
EMA Pain	REF	−0.19 (0.29)	0.505	−0.20 (0.28)	0.466
EMA Fatigue	REF	0.13 (0.35)	0.714	−0.24 (0.34)	0.481
EMA Mood	REF	0.03 (0.28)	0.928	0.14 (0.27)	0.594
EMA Confidence	REF	−0.31 (0.26)	0.238	−0.15 (0.25)	0.568
EMA Concentration	REF	−0.37 (0.29)	0.206	0.02 (0.28)	0.929

Note: regression adjusted for participant age, sex, and race/ethnicity.

**Table 6 jpm-16-00023-t006:** Descriptive Statistics for the System Usability Scale (N = 155).

Measure	Values, n (%)	All Participants (N = 155)	Personalized Trial Arm (N = 81)	Standard-of-Care Arm (N = 74)	*p*-Value ^a^
		Mean (SD; Range)	Mean (SD; Range)	Mean (SD; Range)	
System Usability Scale overall score	155 (73)	81.31 (13.97; 25–100)	82.69 (13.01; 40–100)	79.8 (14.88; 25–100)	0.20
System Usability Scale individual items ^b^					
1. I think that I would like to use this system frequently.	155 (73)	2.59 (1.13; 0–4)	2.67 (1.1; 0–4)	2.51 (1.16; 0–4)	0.40
2. I did not find the system unnecessarily complex. ^c^	155 (73)	3.14 (1.07; 0–4)	3.25 (1.03; 0–4)	3.03 (1.1; 0–4)	0.20
3. I thought the system was easy to use.	155 (73)	3.48 (0.74; 0–4)	3.54 (0.65; 2–4)	3.41 (0.83; 0–4)	0.25
4. I do not think that I would need the support of a technical person to be able to use this system. ^c^	155 (73)	3.70 (0.69; 0–4)	3.78 (0.52; 2–4)	3.62 (0.82; 0–4)	0.17
5. I found the various functions in this system were well integrated.	155 (73)	2.95 (1.01; 0–4)	2.9 (1.02; 0–4)	3 (0.99; 0–4)	0.54
6. I did not think there was too much inconsistency in this system. ^c^	155 (73)	3.14 (1.05; 0–4)	3.14 (1.06; 0–4)	3.15 (1.04; 0–4)	0.94
7. I would imagine that most people would learn to use this system very quickly.	155 (73)	3.26 (0.81; 0–4)	3.36 (0.75; 1–4)	3.16 (0.86; 0–4)	0.13
8. I did not find the system very awkward to use. ^c^	155 (73)	3.25 (0.91; 1–4)	3.32 (0.83; 1–4)	3.18 (0.98; 1–4)	0.33
9. I felt very confident using the system.	155 (73)	3.48 (0.73; 1–4)	3.49 (0.71; 1–4)	3.47 (0.76; 1–4)	0.86
10. I did not need to learn a lot of things before I could get going with this system. ^c^	155 (73)	3.52 (0.80; 0–4)	3.63 (0.77; 0–4)	3.39 (0.82; 1–4)	0.07

^a^ *p*-values come from independent samples *t*-tests between the personalized and standard of care arms. ^b^ Questions rated on a 5-point Likert scale from 0 “Strongly disagree” to 4 to “Strongly agree”. ^c^ Items were initially reverse coded but have been recoded to be on the same scale as other items. The text of these questions has been revised from the original items to reduce confusion.

## Data Availability

Data will be made available on the Open Science Framework (https://osf.io/9mqnp/overview accessed on 9 November 2022) upon publication of the primary outcomes.

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
