# Peer review of "Randomized Personalized Trial for Stress Management Compared to Standard of Care"

_jpm, 2026, doi:10.3390/jpm16010023_

Round 1

Reviewer 1 Report

Comments and Suggestions for Authors

The manuscript “Randomized Personalized Trial for Stress Management Compared to Standard of Care” presents methodologically rigorous investigation into whether personalized N-of-1 trial designs can improve stress management compared with a standard-of-care approach. The study’s concept is innovative and addresses a relevant question in behavioral and personalized medicine. By randomizing 212 participants into two personalized arms and one standard-of-care group, the authors have ensured a robust comparative framework. The design is further strengthened by the use of ecological momentary assessments, integration of objective wearable data from Fitbit devices, and validated measures. The study’s transparency - through preregistration, open-science commitment, and detailed reporting - significantly enhances its scientific credibility.

Despite the null findings regarding the main hypothesis, the work offers valuable insight into the feasibility and challenges of conducting personalized behavioral interventions. The authors report that participants in the personalized arms did not experience significantly greater stress reduction than those receiving standard care, but those who adhered to their personalized recommendations did show a modest benefit. This is a key observation that points to the practical challenges of adherence and engagement. However, while the authors rightly identify low adherence as a limitation, the discussion could have gone further in exploring why personalization did not translate into stronger effects. For example, the manuscript does not fully consider whether the intervention materials themselves - such as the video content - were engaging, easy to use, or sufficiently motivating. It would also be useful to clarify how compliance with the interventions was verified, beyond tracking video access. Given that adherence is central to the interpretation of behavioral outcomes, more detail on how implementation fidelity was controlled would add robustness to the conclusions.

Overall, the study represents a valuable contribution to the field of personalized behavioral research. It demonstrates that large-scale, technology-supported N-of-1 designs are feasible and well accepted, even if methodological refinements are needed to improve their effectiveness. The authors are transparent about the study’s limitations and cautious in interpreting the findings, maintaining scientific balance throughout the discussion.

Author Response

Stress Primary Outcome Paper ([JPM] Manuscript ID: jpm-4002315) 

We thank the editor for their consideration, and the reviewers for their helpful comments on our manuscript. We have carefully considered this feedback and have revised the manuscript accordingly. Below you will find our response to each comment. To ease review of the revised manuscript, changes are indicated by tracked text in the revised manuscript, with line numbers provided below as needed. 

Reviewer 1 

Comments and Suggestions for Authors 

The manuscript “Randomized Personalized Trial for Stress Management Compared to Standard of Care” presents methodologically rigorous investigation into whether personalized N-of-1 trial designs can improve stress management compared with a standard-of-care approach. The study’s concept is innovative and addresses a relevant question in behavioral and personalized medicine. By randomizing 212 participants into two personalized arms and one standard-of-care group, the authors have ensured a robust comparative framework. The design is further strengthened by the use of ecological momentary assessments, integration of objective wearable data from Fitbit devices, and validated measures. The study’s transparency - through preregistration, open-science commitment, and detailed reporting - significantly enhances its scientific credibility. 

Despite the null findings regarding the main hypothesis, the work offers valuable insight into the feasibility and challenges of conducting personalized behavioral interventions. The authors report that participants in the personalized arms did not experience significantly greater stress reduction than those receiving standard care, but those who adhered to their personalized recommendations did show a modest benefit. This is a key observation that points to the practical challenges of adherence and engagement. However, while the authors rightly identify low adherence as a limitation, the discussion could have gone further in exploring why personalization did not translate into stronger effects. For example, the manuscript does not fully consider whether the intervention materials themselves - such as the video content - were engaging, easy to use, or sufficiently motivating. It would also be useful to clarify how compliance with the interventions was verified, beyond tracking video access. Given that adherence is central to the interpretation of behavioral outcomes, more detail on how implementation fidelity was controlled would add robustness to the conclusions. 

RESPONSE: We agree that our discussion could have included additional insights as to why personalization did not transfer into stronger effects. We have added brief discussions regarding the potential impact of intervention materials and aspects of intervention fidelity (lines 482-489).  

Overall, the study represents a valuable contribution to the field of personalized behavioral research. It demonstrates that large-scale, technology-supported N-of-1 designs are feasible and well accepted, even if methodological refinements are needed to improve their effectiveness. The authors are transparent about the study’s limitations and cautious in interpreting the findings, maintaining scientific balance throughout the discussion. 

RESPONSE: Thank you for this feedback. 

Reviewer 2 Report

Comments and Suggestions for Authors

General Evaluation: This manuscript presents a randomized comparison of personalized N-of-1 trials versus standard-of-care for stress management interventions. The topic is timely, relevant, and aligns well with contemporary interests in precision behavioral medicine. The paper is generally well written and clearly structured. However, several areas require clarification, methodological refinement, and stronger interpretation.

  1. Statistical Methodology

The statistical methods are generally appropriate, but several concerns remain:

Primary analysis choice:

The primary comparison relies on a simple two-sample t-test on pre–post change scores. This approach does not fully exploit the richness of EMA longitudinal data and ignores within-person variability. The GLMM analysis mentioned later should arguably be the primary analysis.

Imputation details missing:

The MICE procedure lacks essential information (number of imputations, included variables, convergence checks). These must be reported for reproducibility.

Model specification unclear:

The GLMM description is brief and does not specify the link function (presumably identity), distributional assumptions, whether random slopes were considered, or model diagnostics.

Multiple testing:

Numerous EMA secondary outcomes are tested without adjustment. Although exploratory analyses are acceptable, the risk of Type I error inflation should be acknowledged.

Sensitivity analysis risks:

The significant effect observed only in the “compliers” group may reflect post-hoc subgroup bias. This should be explicitly stated as a limitation rather than framed as support for the intervention.

  1. Sample Description

The sample description is generally adequate, with comprehensive demographic reporting. Still:

Stress levels (PSS ≥ 20) are used as an inclusion criterion, yet baseline PSS scores are not presented in Table 1—this should be added.

There is no overview of baseline EMA stress or other EMA variables, which would help contextualize change scores.

Recruitment strategy heavily involves health system employees; this may limit generalizability and should be noted more explicitly as a limitation.

  1. Tables and Figures

Tables are generally well formatted and clear. Recommendations:

In several tables, the denominators change across analyses (e.g., N=84 vs N=69). Reasons for missingness and unequal denominators should be clarified.

Table 3 and Table 4 overlap conceptually; consider consolidating for clarity.

Figures (SUS Likert plots) are informative but somewhat difficult to interpret due to scale density; consider simpler bar plots or summary statistics.

  1. Adequacy of the Bibliography

The bibliography is rich, relevant, and includes recent sources. A few suggestions:

Add more recent methodological references on N-of-1 trial design and statistical modeling.

Include references on EMA reliability and validity as primary outcome measures.

  1. Study Limitations

The manuscript includes a thoughtful limitations section. However, several important limitations are missing or underdeveloped:

EMA single-item measurement.

A 0–10 single-item stress scale may have limited reliability and sensitivity, especially for detecting between-group differences. This is crucial since EMA stress is the primary outcome.

Potential ceiling/floor effects.

Without presenting baseline stress distributions, it is unclear whether participants had room to improve.

Heterogeneity of stress etiology.

The manuscript mentions not collecting causes of stress, but this is a major limitation: different stressors may respond differently to interventions.

Compliance measurement.

Engagement with intervention videos (duration viewed, frequency) is mentioned but not analyzed. This is a key determinant of treatment effect.

Risk of information overload.

The authors hypothesize this in the Discussion; it should be stated formally in Limitations.

  1. Clinical Implications

Clinical implications are generally described but require strengthening:

The manuscript should clarify that personalized N-of-1 trials may be beneficial only for individuals willing to act on personalized feedback.

The implications for real-world stress management are modest, given the non-significant overall effect.

It would be useful to discuss the cost, scalability, and feasibility of implementing this model in clinical practice.

Author Response

Stress Primary Outcome Paper ([JPM] Manuscript ID: jpm-4002315) 

We thank the editor for their consideration, and the reviewers for their helpful comments on our manuscript. We have carefully considered this feedback and have revised the manuscript accordingly. Below you will find our response to each comment. To ease review of the revised manuscript, changes are indicated by tracked text in the revised manuscript, with line numbers provided below as needed. 

Reviewer 2 

Comments and Suggestions for Authors 

General Evaluation: This manuscript presents a randomized comparison of personalized N-of-1 trials versus standard-of-care for stress management interventions. The topic is timely, relevant, and aligns well with contemporary interests in precision behavioral medicine. The paper is generally well written and clearly structured. However, several areas require clarification, methodological refinement, and stronger interpretation. 

Statistical Methodology 

The statistical methods are generally appropriate, but several concerns remain: 

Primary analysis choice: 

The primary comparison relies on a simple two-sample t-test on pre–post change scores. This approach does not fully exploit the richness of EMA longitudinal data and ignores within-person variability. The GLMM analysis mentioned later should arguably be the primary analysis. 

RESPONSE: This is an excellent point. We chose to make the simple between-group comparison the primary analysis for two reasons.  Firstly, our primary comparison is to identify whether personalized N-of-1 trials help participants to select the best stress management strategy for the follow-up period. Therefore, we wanted to compare stress ratings between the first 2 weeks and the last 2 weeks of the trial. While we could use GLMM to compare the 14 baseline days of EMA data versus the 14 follow-up days, we felt that using a simpler, more interpretable comparison between groups would be easier to report. Secondly, the simple between-groups comparison was the most feasible to base the sample size and power calculations on. The reviewer is correct that the GLMM analyses would provide more nuance but would also be more difficult on which to base our sample size and power calculations. As there are few studies which have attempted to estimate the treatment benefits of personalized N-of-1 trials and no studies which have done so for stress management, calculating power and sample size using a GLMM model would require many assumptions to be made about parameter estimates. Instead, we chose to use assumptions and previous research to generate estimates of the between group effect. Unfortunately, this is why we selected the between-groups comparison as our primary outcome and published the intent to do so, prior to initiating the primary outcome analysis.  

Imputation details missing: 

The MICE procedure lacks essential information (number of imputations, included variables, convergence checks). These must be reported for reproducibility. 

RESPONSE: Thank you for catching this issue. We did not use MICE for the secondary outcomes analysis in this manuscript though we were planning to do so for secondary outcomes analysis in subsequent manuscripts. This language continued forward from previous study documents but was not actually reflected in the analyses we conducted. We’ve revised the manuscript text to state that we explored multiple techniques to deal with missing values in follow-up, including nearest neighbor interpolation and MICE. Finally, we settled on including all individuals with some data during follow-up (ie. 1 or more data points during follow-up). The reason for this is that we were unsure about the proper techniques needed to accurately impute data either within participants or between participants. For individuals missing data during follow-up, we could generate missing values using within-subjects methods such as nearest neighbor interpolation. However, this would mean using stress values from before a participant selected their chosen stress management intervention as values for follow-up. This didn’t seem to be an accurate approach. In addition, we could have used methods such as MICE which would use within-subject variables and between-subject variables to impute missing data. However, our whole design is predicated on focusing on individual differences. The well-established heterogeneity of treatment effects in stress interventions could cause issues with the accuracy of MICE. As a result, we felt that using all available data was the method least subject to bias. 

We intend to fully explore how different imputation methods may inform our primary outcome analyses but felt the scope of this discussion was beyond the scope of the current manuscript.  

Model specification unclear: 

The GLMM description is brief and does not specify the link function (presumably identity), distributional assumptions, whether random slopes were considered, or model diagnostics. 

RESPONSE: Thank you so much for catching this. We actually did not use GLMM models in the current analysis. We were planning to run mixed effects models for additional secondary outcome analyses for future manuscripts. The results in Table 5 depict results from the LME regression analyses. We’ve revised the description in the Table title and in the methods section to more accurately reflect this.  

Multiple testing: 

Numerous EMA secondary outcomes are tested without adjustment. Although exploratory analyses are acceptable, the risk of Type I error inflation should be acknowledged. 

RESPONSE: This is an excellent point. We have added a statement to the results section stating, “These analyses were exploratory and results from these analyses should interpreted cautiously due to potential bias from these multiple comparisons.” (lines 316-318) 

Sensitivity analysis risks: 

The significant effect observed only in the “compliers” group may reflect post-hoc subgroup bias. This should be explicitly stated as a limitation rather than framed as support for the intervention. 

RESPONSE: This is a good point. We have explicitly stated in the discussion this sensitivity analysis was not identified a priori and that: “This sensitivity analysis may be biased.” (line 447) 

Sample Description 

The sample description is generally adequate, with comprehensive demographic reporting. Still: 

Stress levels (PSS ≥ 20) are used as an inclusion criterion, yet baseline PSS scores are not presented in Table 1—this should be added. 

RESPONSE: Thank you for this suggestion. We have added baseline PSS scores and baseline EMA Stress scores to Table 1. No significant differences in baseline stress were identified between the personalized and standard-of-care arms.  

There is no overview of baseline EMA stress or other EMA variables, which would help contextualize change scores. 

RESPONSE: Thank you for pointing this out; we have added a brief overview of how EMA stress was assessed and scored, as well as for the other EMA variables (e.g., fatigue) to help the reader contextualize the results (lines 212-216 and 236-238).  

Recruitment strategy heavily involves health system employees; this may limit generalizability and should be noted more explicitly as a limitation. 

RESPONSE: Thank you for pointing this out. While we also had extensive remote recruitment strategies, such as via Facebook, we have added a sentence to address this in the limitations section (lines 579-581). We have also expanded the Methods section under Participants, Recruitment and Consent section to emphasize additional screening sources outside of Northwell (lines 114-119). 

Tables and Figures 

Tables are generally well formatted and clear. Recommendations: 

In several tables, the denominators change across analyses (e.g., N=84 vs N=69). Reasons for missingness and unequal denominators should be clarified. 

RESPONSE: For some analyses, the sample size differed from the full sample. For example, in Table 2 we report change in stress and other outcomes between follow-up and baseline; here a footnote indicates “Smaller sample sizes existed for this comparison N=69 (personalized) and 60 (standard)”, which applies to the Weekly Stress outcome in particular. The N in Table 3 and Table 4 (N=81 in Arms 1 and 2 and N=73 in Standard of Care Arm) differs from the N in Table 2 because Tables 3 and 4 report findings based on treatment selection following the completion of the trial. Table 2 reports change in the primary outcome.      

Table 3 and Table 4 overlap conceptually; consider consolidating for clarity. 

RESPONSE: Thank you for this suggestion. Prior to initial submission we discussed consolidating Table 3 and Table 4 but ultimately decided the results would be clearer if presented separately. Table 3 shows the differences in treatment selection and recommendation by study arm. The goal of this table is to identify whether selection and recommendation differ by intervention assignment. Table 4 ignores study arm and instead focuses the concordance between recommended interventions and selected interventions solely in the personalized trial arms. This table is used to identify whether participants in personalized N-of-1 trials chose to follow the recommendations in their reports. Because of the differences in inference which can be drawn from each table, we chose to keep them separate. 

Figures (SUS Likert plots) are informative but somewhat difficult to interpret due to scale density; consider simpler bar plots or summary statistics. 

RESPONSE:  We agree with the reviewer that this is visually very busy. However, we prefer these graphs because of the density of information which can be shown. We can see not only the proportion of individuals who endorsed each response for each item but also the distribution of those responses across items. For example, it’s clear that participants responded heterogeneously regarding their feelings about using the system frequently and that this heterogeneity was maintained in personalized and standard-of-care arms. We have added a key at the bottom of the figure to improve interpretation.  

Adequacy of the Bibliography 

The bibliography is rich, relevant, and includes recent sources. A few suggestions: 

Add more recent methodological references on N-of-1 trial design and statistical modeling. 

RESPONSE: Thank you for this suggestion; we have added N=20 references throughout the manuscript to update our bibliography. These citations are reflected in the tracked changes so it’s clear where they have been added and what statements they are supporting. This includes addressing the editor’s concern regarding the proportion of pre-2017 references; all but one of our added references is within the last 8 years.   

Include references on EMA reliability and validity as primary outcome measures. 

RESPONSE: Thank you for this suggestion; we have added a few references for EMA measures in the Methods (lines 109; 209) and Discussion (line 571) sections. 

Study Limitations 

The manuscript includes a thoughtful limitations section. However, several important limitations are missing or underdeveloped: 

EMA single-item measurement. A 0–10 single-item stress scale may have limited reliability and sensitivity, especially for detecting between-group differences. This is crucial since EMA stress is the primary outcome. 

RESPONSE: While the EMA stress scale is a validated measure, that is also well-suited to a fully remote digital trial and has previously been utilized as a self-reported stress measure in other personalized N-of-1 trials, we understand this concern. We have added a sentence in the limitations to address this limitation (lines 575-579).  

Potential ceiling/floor effects. Without presenting baseline stress distributions, it is unclear whether participants had room to improve. 

RESPONSE: Thank you for pointing this out. We have added baseline stress scores (PSS) and baseline EMA stress scores to Table 1, so that when the reader views Table 2, there is a basis for interpreting room for improvement. We also added the PSS and EMA Stress mean(SD) in the results section (lines 325-327). 

Heterogeneity of stress etiology. The manuscript mentions not collecting causes of stress, but this is a major limitation: different stressors may respond differently to interventions. 

RESPONSE: We agree this is a valid concern.  Our limitations section states:  

“Another potential limitation is that individuals may have had different sources and  causes of stress. Although perceived stress is a complex and dynamic  phenomenon influenced by both modifiable (e.g., coping skills) and non-modifiable  (e.g., environment) factors [46,47], to make the content and the design of the  intervention broadly applicable, we did not identify causes of stress or exclude  individuals with particular causes of stress. However, this may have resulted in  enrolling some participants with variable or potentially non-modifiable causes of  stress.”  

We have expanded the discussion on this limitation by adding “Because different causes of stress may respond differently to interventions, this is an important consideration for future work” (lines 569-570) 

Compliance measurement. Engagement with intervention videos (duration viewed, frequency) is mentioned but not analyzed. This is a key determinant of treatment effect. 

RESPONSE: The reviewer is correct that discussion of the engagement with the intervention videos is somewhat limited. This is because the primary outcome is to analyze the relative effectiveness of personalized N-of-1 trials versus standard-of-care for stress management. We agree with the reviewer that treatment engagement is an important moderator of our intervention effect. However, the variables quantifying engagement (e.g. time of viewing, duration of viewing, and frequency of viewing) are complex and do not map well universally across arms. For example, individuals in the personalized arms will only be watching 1 stress management video during a given period while individuals in the standard-of-care arm are allowed to watch all videos at once. Examining the effect of engagement on the intervention requires a thoughtful, in-depth analysis which we believe is beyond the scope of this primary outcome paper. In addition, we did not have a priori analyses defined or a statistical plan specified to assess these engagement metrics. We do agree this could be the basis for an exciting secondary analysis. 

Risk of information overload. The authors hypothesize this in the Discussion; it should be stated formally in Limitations. 

RESPONSE: We agree with the reviewer that information overload may be a potential limitation for some of the participants in our study. However, we are not sure this is universally the case. Some participants, particularly those who followed the recommendations in the personalized trial report, may have not been overloaded by the information they received. It is also possible that individuals who did not follow recommendations may not have felt information overload. For example, an individual may have used the large personalized data in their report to determine that the stress reduction benefits of yoga were outweighed by a potential increase in pain. Therefore, we believe information overload should be talked about in the discussion as more research is required to determine whether this was truly a detriment in the current trial.  

Clinical Implications 

Clinical implications are generally described but require strengthening: 

The manuscript should clarify that personalized N-of-1 trials may be beneficial only for individuals willing to act on personalized feedback. 

RESPONSE: Although we have stated that “personalized N-of-1 trials may be beneficial for participants but only for the sub-sample of individuals who followed the recommendations in their personalized report” (lines 443-445), we have expanded on this point to further clarify that personalized N-of-1 trials may be beneficial only for individuals willing to act on personalized feedback (lines 531-532).  

The implications for real-world stress management are modest, given the non-significant overall effect. 

RESPONSE: We agree that more work needs to be done to determine factors that may influence an individual’s decision to follow a personalized report to manage their stress. For example, in future trials we could refine how personalized N-of-1 trials are conducted, how participants interact with the intervention components, and how trial results are subsequently reported to participants, in order to increase uptake of the stress management recommendations.  

It would be useful to discuss the cost, scalability, and feasibility of implementing this model in clinical practice. 

RESPONSE: We understand the importance of including implications for clinical practice for the journal’s readership. Our study design, being fully remote, digital, and using existing technologies, has some inherent advantages for cost, scalability, and feasibility. For example, leveraging digital recruitment and pre-recorded interventions delivered via widely accessible platforms demonstrates significant potential for scalability to broader populations, overcoming geographical barriers inherent in traditional care. From a feasibility standpoint, our study's highly positive system usability scores suggest acceptability among users. While initial cost will be inevitable, future cost reduction is possible through further automation of report generation. Challenges that may still remain in integrating data collection and personalized feedback into existing clinical workflows include ensuring adequate clinician training and buy-in, and, as our findings suggest, effectively communicating personalized recommendations to maximize patient uptake and behavioral change. We have summarized these points and added a brief discussion on clinical implications in the conclusion (lines 584-592).  

Round 2

Reviewer 2 Report

Comments and Suggestions for Authors

I sincerely appreciate the effort you have put into revising and improving the article; your work has clearly strengthened the manuscript and enhanced its overall clarity and rigor.

Author Response

Thanks for your comments.